# Diagnosis and Management of Pulmonary Hypertension: New Insights

**DOI:** 10.3390/diagnostics14182052

**Published:** 2024-09-16

**Authors:** Despoina Ntiloudi, Nearchos Kasinos, Alkistis Kalesi, Georgios Vagenakis, Anastasios Theodosis-Georgilas, Spyridon Rammos

**Affiliations:** 1Department of Cardiology, Tzaneio General Hospital of Piraeus, 18536 Piraeus, Greece; ntiloudid@gmail.com (D.N.); nearchoskassinos@gmail.com (N.K.); eileen_calessis@yahoo.gr (A.K.); medtheodosis@hotmail.com (A.T.-G.); 2Echocardiography Training Center of Tzaneio ‘D. Beldekos’, 18536 Piraeus, Greece; 3Department of Pediatric Cardiology and Adult Congenital Heart Disease, “Onassis” Cardiac Surgery Center, 17674 Athens, Greece; vagenakg@yahoo.co.uk

**Keywords:** pulmonary arterial hypertension, pulmonary hypertension, chronic thromboembolic pulmonary hypertension, diagnosis, risk assessment, therapy

## Abstract

Over the last decades, significant progress has been achieved in the pulmonary hypertension (PH) field. Pathophysiology of PH has been studied, leading to the classification of PH patients into five groups, while the hemodynamic definition has been recently revised. A diagnostic algorithm has been established and awareness has been raised in order to minimize diagnosis delay. The pulmonary arterial hypertension (PAH) treatment strategy includes the established three pathways of endothelin, nitric oxide-phosphodiesterase inhibitor, and prostacyclin pathway, but new therapeutic options are now being tested. The aim of this review is to summarize the existing practice and to highlight the novelties in the field of PH.

## 1. Introduction

Pulmonary hypertension (PH) is a severe condition characterized by elevated mean pulmonary arterial pressure (mean PAP) and can be encountered in a wide variety of diseases, such as cardiovascular, pneumonological, and rheumatological, among others [1]. After the two large cohort studies by Maron et al., which showed that patients with borderline mean PAP (19–24 mmHg) and pulmonary vascular resistance (PVR) of more than 2.2 Wood Units (WU) have increased all-cause mortality and hospitalization, the definition of PH has changed in the current guidelines [2,3]. PH is defined by a mean PAP > 20 mmHg at rest, estimated with right heart catheterization (RHC). Precapillary PH is defined by mean PAP > 20 mmHg, PVR > 2 WU, and pulmonary arterial wedge pressure (PAWP) < 15 mmHg, while postcapillary PH is defined by mean PAP > 20 mmHg and PAWP > 15 mmHg [1]. In the latter group, the precapillary component is present when PVR > 2 WU (combined PH) but otherwise is deemed isolated postcapillary PH (PVR < 2 WU). The diastolic pressure gradient is no longer used for this classification, while exercise PH was introduced as mean PAP divided by cardiac output (slope) between rest and exercise at more than 3 mmHg/L/min [1,4].

The basic structure of clinical classification has been kept [1]. Group 1, the pulmonary arterial hypertension (PAH) group, includes idiopathic (responders and no responders to vasoreactivity test) and heritable, as well as PAH associated with drugs and toxins, connective tissue disease, congenital heart disease, human immunodeficiency virus, portal hypertension and schistosomiasis, PAH with venous/capillary component, and persistent PH of the newborn [1,4]. Group 1 is very rare with an estimated prevalence of 15–50 persons per million [5]. In the past, it was thought that PAH affected young women. However, contemporary data from registries have shown that PAH is now frequently diagnosed in older patients, even individuals more than 65 years old with cardiovascular comorbidities [6].

Group 2 consists of patients with left heart disease complicated with PH, isolated postcapillary, or combined with a precapillary component. PH is the outcome of the increased left-sided filling pressures. It is a very common complication among patients with heart failure (HF) with preserved, reduced, or mildly reduced ejection fraction, as well as among patients with valvular disease or congenital/acquired cardiovascular conditions leading to postcapillary PH [7,8]. Given that the number of patients with HF has dramatically increased, the prevalence of PH associated with left heart disease will be further increased.

Group 3, PH associated with lung disease and/or hypoxia, is a common complication in patients with parenchymal/interstitial pulmonary disease [9]. It is usually mild and the prevalence of PH depends on the severity of the disease and is associated with worse functional class and outcomes [10,11]. PH associated with pulmonary artery obstruction constitutes Group 4 and is mainly represented by chronic thromboembolic PH (CTEPH). The prevalence is 26–38 cases per million adults and about 75% have a history of acute pulmonary embolism [12,13,14]. Finally, PH with unclear or multifactorial mechanisms are classified in Group 5.

In general, PH affects 1% of the general population and even though the therapeutic progress has resulted in better survival rates, the prognosis of PH patients is still poor [6]. The purpose of this review is to provide an update on diagnosis, risk stratification, and management of PH.

## 2. Diagnosis

Despite the significant progress in the PH field, there is still an urgent need for even more on-time diagnosis and fast-track referral to expert PH centers when PAH or CTEPH is suspected [1,15]. The initial diagnosis assessment is also important to clarify the underlying cause of PH and to detect comorbidities and complications of PH.

The first presenting symptoms include dyspnea on exertion, fatigue, and rapid exhaustion [15,16]. Palpitations, hemoptysis, syncope, and symptoms due to pulmonary artery dilatation (chest pain, hoarseness, and symptoms by the compression of the bronchi) may also be present. Clinical signs in patients with PH can be classified as signs of PH (cyanosis, loud second heart sound, systolic murmur of tricuspid regurgitation, diastolic murmur of pulmonary regurgitation), signs of the right ventricle (RV) backward failure (peripheral edema, distended jugular veins, ascites, hepatomegaly, and abdominal distention), signs pointing towards the underlying cause of PH, as well as signs of RV forward failure (cyanosis, pallor, cool extremities, and prolonged capillary refill) [1].

An electrocardiogram may raise suspicion of PAH when right ventricular strain and right axis deviation are present [17,18]. It can also detect arrhythmias or signs of left heart disease. Chest X-ray could be helpful with the underlying cause of PH, for example in Group 3 patients [1]. The most typical findings are right atrium/RV/pulmonary artery enlargement with peripheral pulmonary vasculature pruning. Initial blood work-up should include brain natriuretic peptide (BNP)/N-terminal pro-BNP (NT-proBNP), basic immunological laboratory work-up, as well as screening for antiphospholipid syndrome in patients with CTEPH [15]. Pulmonary functional tests with lung diffusion capacity for carbon monoxide and arterial blood gases are also of high importance in the initial work-up [19]. A cardiopulmonary exercise test (CPET) is useful to assess the mechanism and the severity of exercise intolerance and to estimate the prognosis in patients with PH [20,21,22]. Typical findings in PH patients are low peak oxygen uptake and high ventilatory equivalent for carbon dioxide [20].

Echocardiography is a valuable noninvasive screening tool [23,24]. Information on both right and left ventricles can be obtained by echocardiography. Given that PH associated with left heart disease is the most common type of PH, echocardiography can help further distinguish these patients and estimate the likelihood of diastolic dysfunction [25]. The thresholds regarding tricuspid regurgitation velocity (TRV) have not changed, despite the revised hemodynamic definition [1,26]. A peak TRV > 3.4 m/s suggests a high probability of PH, while TRV 2.9–3.4 suggests an intermediate probability and additional signs suggestive of PH should be taken into account. More specifically, the signs from the ventricles are right ventricle/left ventricle basal diameter/area ratio of more than 1 and flattening of the interventricular septum (left ventricular eccentricity index >1.1 in systole and/or diastole). The recently added parameter that reflects RV-pulmonary artery coupling is tricuspid annular plane systolic excursion/systolic pulmonary arterial pressure ratio <0.55 mm/mmHg [1]. Signs from the pulmonary artery are right ventricular outflow tract acceleration time <105 ms and/or mid-systolic notching, early diastolic pulmonary regurgitation velocity >2.2 m/s, and pulmonary artery diameter >25 mm or more than aortic root diameter. Finally, additional signs are right atrium area (measured in end-systole) >18 cm^2^ and dilated inferior vena cava (>21 mm with decreased inspiratory collapse).

A ventilation/perfusion lung scan is required in the diagnostic work-up to rule out CTEPH [1,27]. Nonmatched perfusion defects, except for CTEPH, may be rarely observed in idiopathic/hereditary PAH or pulmonary veno-occlusive disease [28]. In chest computed tomography (CT), signs of PH, such as main pulmonary artery dilatation and right atrial/ventricular dilatation, can be detected [29,30]. Also, CT can be helpful in the delineation of PH etiology, especially for Group 3 patients, as well as for veno-occlusive disease, where septal lines, lymphadenopathy, and centrilobular ground-glass opacities are observed [1]. Webs or bands in pulmonary arteries in CT pulmonary angiopathy are suggestive of CTEPH [31]. Cardiac magnetic resonance could be used rarely during the initial diagnostic work-up but can be helpful in cases with suspected congenital heart disease [16,32].

Early diagnosis is of major importance in PH patients. Therefore, in high-risk patients screening for PH is essential. Especially in patients with scleroderma, an algorithm named DETECT has been developed to identify asymptomatic patients with PAH [33]. In this algorithm, step 1 includes 6 clinical variables, while step 2 includes echocardiographic variables and then, according to the outcome, the patient is referred for RHC.

RHC is the gold standard for the diagnosis and classification of PH [1]. It must be performed in specialized centers with standardized protocols in order to report a full set of hemodynamics. In a multicenter study with 7.218 procedures, complications were reported in about 1.1% and usually were related to venous access, while procedure-related mortality was reported in 0.055% [34]. Optimized patient volume and zeroing in the mid-thoracic level are of high importance for the accuracy of the RHC measurements [35]. All measurements should be performed at end-expiration; stepwise assessment of oxygen saturation should be performed when a shunt is suspected and in case of a shunt, the Fick method should be used for cardiac output measurement [35,36]. Vasoreactivity testing is performed in patients with idiopathic, heritable, or drug-induced PH in order to identify acute vasoresponders, who can be treated with high-dose calcium channel blockers [1,37]. This test is performed mainly with inhaled nitric oxide or inhaled iloprost [1]. A positive acute response is defined as a reduction in mPAP by ≥10 mmHg to reach an absolute value ≤40 mmHg, with increased or unchanged CO [4]. In patients with a clinical phenotype of HF with preserved ejection fraction and a PAWP < 15 mmHg in RHC, a fluid challenge may be helpful to reveal the increase in PAWP ≥ 18 mmHg when a rapid infusion of 500ml saline is injected [38,39].

## 3. Management

### 3.1. Pulmonary Arterial Hypertension

The treatment goal for PH patients is to achieve and maintain a low-risk profile [1]. A multiparameter risk stratification approach is proposed for these patients and there are several validated risk tools, including the Swedish Pulmonary Arterial Hypertension Registry, the Comparative, Prospective Registry of Newly Initiated Therapies for PH (COMPERA), the French PH Network Registry, and the US Registry to Evaluate Early and Long-term PAH Disease Management (REVEAL) risk equation [40,41,42,43,44,45,46]. In the previous European guidelines on PH, a three-strata model was proposed but the majority of the patients were stratified in the intermediate risk. Therefore, the risk stratification model currently proposed by the European guidelines for the initial assessment consisted of 3 strata (the estimated 1-year mortality in low risk is less than 5%, in the intermediate risk is 5–20%, and in high risk more than 20%). For the follow-up assessment, a 4 strata model has been introduced in order to better discriminate the intermediate group and guide accordingly the therapeutic strategy (the observed 1-year mortality was 0–3%, 2–7%, 9–19%, and more than 20% for the low, intermediate-low, intermediate-high, and high-risk group, respectively) [1,40,47]. In the initial assessment, a full set of data is taken into account, including signs of right HF, progression of symptoms, history of syncope, WHO functional class, 6-min walking distance (MWD), CPET parameters, BNP or NT-proBNP, echocardiography, cardiac magnetic resonance, and hemodynamic parameters [1]. In the follow-up assessment, only WHO functional class, 6 MWD, and BNP/NT-proBNP are included in the model, while additional parameters can be collected per individual case (Figure 1) [1]. However, it should be noted that age, sex, PAH type, comorbidities such as renal insufficiency, diabetes mellitus, etc., as well as major complications, i.e., hemoptysis, arrhythmias, and pulmonary dilatation that compresses left main coronary artery are not included in these risk assessment tools and should be considered on an individual basis [48].

There are general measures that should be followed for patients with PAH (Figure 2). Physical activity within symptom limits is encouraged, while supervised rehabilitation programs are recommended in clinically stable patients, who are on optimal medical treatment [49,50]. Immunization against influenza, SARS-CoV-2, and pneumococcus is recommended [1]. Long-term oxygen therapy is advised when PaO_2_ < 8 kPa (60 mmHg) [1]. Diuretics are recommended in patients with right HF and fluid retention [51], while data on anticoagulation are conflicting and therefore an individual case-by-case decision should be taken [52]. Drugs effective in left HF (such as angiotensin-converting enzyme inhibitors, angiotensin receptor blockers, angiotensin receptor–neprilysin inhibitors, sodium-glucose cotransporter-2 inhibitors, beta-blockers, or ivabradine) are not recommended in patients with PAH [1]. In patients with iron deficiency and anemia, iron supplementation improved right ventricular function and exercise tolerance [53].

In patients with positive acute vasoreactivity testing, high doses of calcium channel blockers are recommended [37,54]. Complete reassessment, including RHC, should be performed 3 months after the initiation of the treatment to assess its efficacy and safety. Patients are recommended to continue in high doses of calcium channel blocker when a marked hemodynamic improvement is noted, with mean PAP < 30 mmHg and PVR < 4 WU, and the patients are asymptomatic or oligosymptomatic (NYHA class I or II) [1]. Otherwise, targeted PAH treatment is recommended to be initiated. Side effects of high doses of calcium channel blockers are hypotension and peripheral edema.

Regarding PAH drugs, the three current therapeutic targets are the endothelin pathway, the nitric oxide (NO)-soluble guanylate cyclase (sGC)-cyclic guanosine monophosphate (cGMP) pathway, and the prostacyclin pathway [55]. In the endothelin pathway, the three endothelin receptor antagonists (ERAs) are ambrisentan, bosentan, and macitentan. The endothelin receptor antagonists have favorable effects on exercise capacity, symptoms, hemodynamics, and time from clinical worsening [56,57,58]. The approved doses in adults for ambrisentan are 5 or 10 mg once daily, bosentan 125 mg twice daily, and macitentan 10 mg once daily. The major adverse effects are peripheral edema for ambrisentan, abnormal liver function for bosentan, and anemia for macitentan [1]. Abnormal liver function in patients treated with bosentan occurs around one-tenth, and therefore liver function testing should be performed once monthly [59].

In the NO/cGMP pathway belong phosphodiesterase type 5 inhibitors (PDE5i), sildenafil and tadalafil, which inhibit the degradation of cGMP, and a guanylic cyclase stimulator (GCs), riociguat, that enhances cGMP production. Both PDE5i and riociguat have favorable effects on exercise capacity, symptoms, hemodynamics, and time to clinical worsening [60,61,62,63]. The approved dose of sildenafil is 20 mg three times daily, of tadalafil 40 mg once daily, and of riociguat the starting dose is 1 mg three times daily with titration to 2.5 mg three times daily. The side effects of these drugs are mainly headache, flushing, and epistaxis [61].

In the prostacyclin pathway belong the prostacyclin analogs (epoprostenol, iloprost, treprostinil, beraprost) and selexipag, a prostacyclin receptor agonist [64]. Epoprostenol has a very short half-time and therefore needs continuous IV administration via an infusion pump, but has shown a reduction in symptoms and was the first drug that showed a reduction in mortality [65]. Given the method of delivery, except for the general side effects that this category of drugs has (headache, flushing, jaw pain, and diarrhea), patients under epoprostenol might present with pump malfunction, local site infection, catheter obstruction, and sepsis [66]. Iloprost requires six to nine inhalations, while treprostinil is usually subcutaneously administrated, but almost one-tenth of patients discontinue the drug because of site pain [67,68]. Both these drugs have improved exercise capacity and hemodynamics. Selexipag is orally administered, usually on top of double oral therapy [69]. Ralinepag is also an oral prostacyclin receptor agonist with a longer half-life, which has shown promising results, and therefore the results from the phase III trial are anticipated [70].

The treatment algorithm has changed in the initial assessment. Currently, the presence of comorbidities defines the treatment strategy, as well as the follow-up, since the risk stratification model consists of 4 strata [1]. If the patient presents with cardiopulmonary comorbidities, monotherapy with ERAs or PDEi should be considered (Class IIa) and an individual approach is proposed as a follow-up [1,71]. In patients without cardiopulmonary comorbidities, the initial strategy depends on the three strata risk stratification model [1]. For those considered low or intermediate risk, an initial combination therapy is recommended (Class I). This was firstly supported by the AMBITION trial, where initial combination therapy with tadalafil and ambricentan showed better results in the first clinical failure event (primary outcome) compared to initial monotherapy [72,73]. For those considered high risk, an initial triple combination therapy, including intravenous or subcutaneous prostacyclin analog, should be considered (Class IIa) [74]. In the follow-up assessment, where the 4 strata model is taken into account, those in low risk are recommended to follow the same treatment (Class I) as those in intermediate-low risk. Adding selexipag should be considered with a class of indication IIa of switching PDEi to sGC with an indication IIb [75,76,77]. For those in the follow-up who are stratified in the intermediate-high or high risk, adding an intravenous or subcutaneous prostacyclin analog and/or evaluation for lung transplantation should be considered (class IIa) [78].

Beyond the three target pathways, novel therapies are currently tested in randomized control trials. The most popular are the transforming growth factor-β (TGF-β) pathway and platelet-derived growth factor (PDGF) pathway [79]. Sotatercept suppresses TGF-β and enhances bone morphogenetic protein receptor 2, with the purpose of helping in cell apoptosis and endothelium function [80]. PULSAR and STELLAR trials have shown favorable effects on functional class, NT-proBNP, and hemodynamics, and therefore sotatercept was approved by the Food And Drug Administration for treating patients with PAH [81,82,83]. Side effects are an increase in hemoglobin, thrombocytopenia, telangiectasia, epistaxis, and dizziness [79]. Imatinib is a tyrosine kinase inhibitor, which targets the PDGF pathway, by inhibiting the proliferation of pulmonary artery smooth muscles [84]. In the IMPRES trial, imatinib showed improvement in 6MWD but had serious adverse events, mainly subdural hematoma, and a high rate of discontinuation was observed therefore alternative formulations of imatinib and other drugs of the PDGF pathway are being tested [85,86].

Interventional therapies, such as atrial septostomy, Potts shunt, and pulmonary artery denervation are rarely used, given the little data from randomized control trials [1]. With the first two techniques, the right ventricle is decompressed and the cardiac index is increased in the cost of desaturation, while the latter one is based on the inhibition of the baroreflex, which is associated with vasoconstriction [87,88,89].

Regarding patients with shunt lesions, management varies according to the type of shunt and the PVR [90]. In patients with PVR < 3 WU, shunt closure is recommended (Class I), while for those with PVR 3–5, WU, shunt closure should be considered (Class IIa). In patients with atrial septal defect and PVR > 5 WU that declines to 3–5 WU with PAH drugs, or with post-tricuspid shunt and PVR > 5 WU, shunt closure may be considered (IIb). For those with atrial septal defect and PVR > 5 WU despite the administration of PAH drugs, a shunt is not recommended to be closed (Class III). In pediatric patients, the PVR index is taken into account and is generally considered safe to close a shunt with a PVR index <4 WU and PVR index/systemic vascular resistance index <0.5, with evidence of reactivity with acute vasoreactivity testing [91].

### 3.2. Pulmonary Hypertension Associated with Left Heart Disease

The primary goal of the treatment strategy in these patients is to optimize the treatment of the underlying condition, including both established medical and interventional therapies [8]. In general, routine use of PAH drugs in this category of patients is not recommended, since most trials did not have positive results [92,93,94]. However, an individual approach should be followed in patients with severe precapillary components (PVR > 5 WU) [1]. Regarding patients with advanced HF with reduced ejection fraction (EF), implantation of a left ventricular assist device has shown a significant reduction in mean PAP, while there are no randomized control trials to support the use of PAH drugs [95]. For patients with HF with preserved EF, sildenafil has shown some favorable effects in patients with a severe precapillary component and therefore in the recent guidelines for PH, no recommendation was given for or against the use of sildenafil in patients with HF with preserved EF and combined post and precapillary component. However, PDE5i is not recommended for isolated postcapillary PH [1,96]. Regarding patients with persistent PH after successful heart valve replacement or repair, the SIOVAC trial showed that the use of sildenafil in this context was associated with an increased risk of deterioration and death [97].

### 3.3. Pulmonary Hypertension Associated with Lung Disease and/or Hypoxia

Likewise, for patients with PH associated with left heart disease also in this group of patients, treating the underlying cause is the main goal of the treatment strategy [98]. Current data do not support the use of PAH drugs. However, in those patients with lung disease and severe PH, a referral to a PH center and an individual approach is recommended [1]. Regarding patients with PH and interstitial lung disease, given the positive findings from the INCREASE trial, inhaled treprostinil has an indication of IIb in these patients [99].

### 3.4. Chronic Thromboembolic Pulmonary Hypertension

The management strategy of patients with CTEPH depends on the anatomical distribution of the lesions as proximal, distal, and microvascular, and includes pulmonary endarterectomy (PEA), balloon pulmonary angioplasty (BPA), and PAH drugs [100]. All patients are recommended to receive life-long therapeutic anticoagulation [1]. Even though the data on the use of direct oral anticoagulation (DOAC) therapy in this group of patients is limited, an increasing use of these agents has been reported. Data from the EXPERT study by Humber M et al. showed similar hemorrhagic events in the DOAC and vitamin K antagonist (VKA) group, but the thrombotic and/or embolic event rate was higher in the DOAC group [101]. In the recent guidelines, experts are in favor of using VKA antagonists, and in the case of antiphospholipid syndrome, which is detected in 10% of CTEPH cases, VKAs are recommended (Class I) [1].

PEA is the treatment of choice in operable patients since it was shown to significantly lower mortality [102]. Characteristics that predict good long-term outcomes are a history of deep vein thrombosis/pulmonary embolism, no signs of right HF, absence of comorbidities, NYHA class II or III, disease concordant on all images, bilateral lower lobe disease, and favorable hemodynamics, including PVR < 1000 dyn·s·cm^−5^, in proportion to size and number of obstructions on imaging and higher PA pulse pressure [100]. Operability is mainly dependent on team experience, accessibility of obstruction lesions, comorbidities, and severity of PH [103]. The insertion of the inferior vena cava filter device prior to the PEA did not influence long-term survival and therefore is not indicated [100]. Even though PEA has changed the landscape for these patients, one-fourth may present with postoperative PH [104].

In these cases, with persistent/recurrent PH after PEA or in patients ineligible for PEA, an interventional procedure, i.e., balloon pulmonary angioplasty (BPA) is recommended [1]. BPA has shown favorable effects on hemodynamics, functional capacity, and function of RV, especially in expert centers [105]. A staged procedure should be followed in a high-volume center, in order to minimize the complications [1]. Complications include wire perforation resulting in vascular injury and lung injury presenting with hypoxia and/or hemoptysis. Pretreatment should be considered prior to the BPA procedure, especially in patients with PVR > 4 WU (IIa indication) [106].

In patients with microvascular components and inoperable CTEPH of recurrent/persistent PH after PEA, PAH drugs are indicated. Based on the CHEST trials, riociguat has a level of evidence I in symptomatic patients of this category, while subcutaneous treprostinil has an indication IIb [63,107]. It should be noted, however, that combination therapy is used in daily clinical practice and there are some data, for example MERIT-1 trial, to support this practice [108].

### 3.5. Pulmonary Hypertension with Unclear and/or Multifactorial Mechanisms

Treatment of the underlying cause is also the primary goal in this category of patients [1]. The off-label use of PAH drugs should be performed with caution, since in the underlying mechanism a venous pulmonary component might be present.

## 4. Conclusions

Even though significant progress has been accomplished in the diagnosis and treatment of patients with PH, PH awareness should still be raised to succeed in earlier recognition of the disease. More multicenter randomized trials should be designed and performed, especially for groups of patients for whom evidence for use of PAH drugs is limited.

## Figures and Tables

**Figure 1 diagnostics-14-02052-f001:**
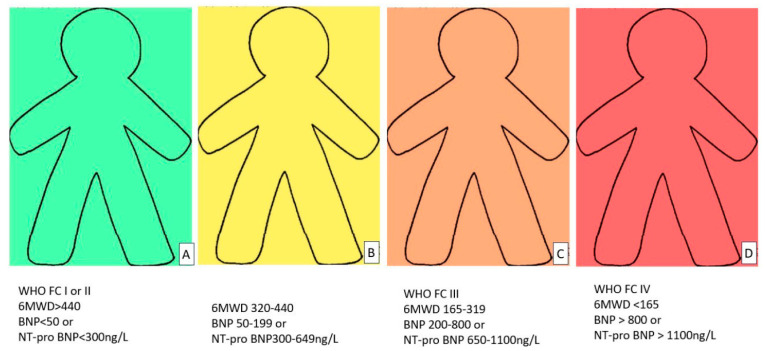
Four strata models (green-low (**A**), yellow-intermediate low (**B**), orange-intermediate high (**C**), and red-high risk (**D**)) for the risk assessment in the follow-up. 6MWD: 6-min walking distance, BNP: brain natriuretic peptide, NT-proBNP: N-terminal prohormone of brain natriuretic peptide, and WHO FC: World Health Organization functional class.

**Figure 2 diagnostics-14-02052-f002:**
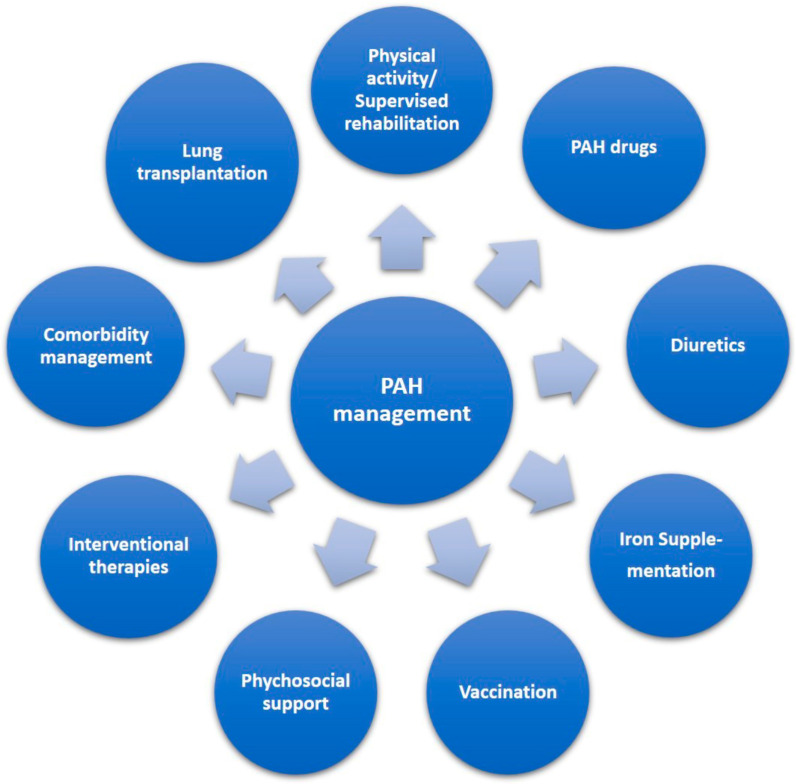
The pillars of pulmonary arterial hypertension (PAH) management.

## Data Availability

Not applicable.

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
