# Peer review of "Diagnosis and Management of Pulmonary Hypertension: New Insights"

_diagnostics, 2024, doi:10.3390/diagnostics14182052_

Round 1
Reviewer 1 Report
Comments and Suggestions for Authors This manuscript is devoted to an important problem of modern cardiology - the diagnosis and treatment of pulmonary hypertension. A careful analysis of the review article left a very positive impression of the work done by the authors: a large number of literature sources were analyzed; current data on the problem were presented in accordance with the latest guidelines of the European Society of Cardiology (ESC) and the European Respiratory Society (ERS) for the management of pulmonary hypertension. However, I would like to point out to the authors a number of minor shortcomings and suggest options for eliminating them: 1. Databases and keywords for finding sources on the problem of interest to the authors are not indicated. 2. References: 34 out of 109 sources are older than 10 years (30%). However, the use of such early publications can be justified by an evolutionary approach to describing classification issues, risk stratification models and the constant emergence of new drugs in the treatment of pulmonary hypertension. 3. The title of Figure 2 states “Basics of Management of Pulmonary Hypertension (PH),” and the figure description in the text provides general measures that should be followed in patients with pulmonary arterial hypertension. The terms pulmonary hypertension and pulmonary arterial hypertension are not identical. 4. A small number of keywords reflecting the content of the article; their expansion will help increase the citation of this work. 5. The manuscript uses many abbreviations. I recommend to provide a list of abbreviations.Author Response
Please see the attachment

Reviewer 2 Report
Comments and Suggestions for Authors
This is a review article on pulmonary hypertension. The authors describes diagnosis and management of the disease. They report the latest knowledge of the disease and referred many clinical trials. I think this manuscript is worthwhile to be published in the journal. I have a minor comment.
P 7, L312: PEA should be Pulmonary endarterectomy (PEA)?
